# Peening Techniques for Surface Modification: Processes, Properties, and Applications

**DOI:** 10.3390/ma14143841

**Published:** 2021-07-09

**Authors:** Merbin John, Prasad Rao Kalvala, Manoranjan Misra, Pradeep L. Menezes

**Affiliations:** 1Department of Mechanical Engineering, University of Nevada, Reno, NV 89557, USA; merbinjohn@nevada.unr.edu; 2Department of Chemical and Materials Engineering, University of Nevada, Reno, NV 89557, USA; pkalvala@unr.edu (P.R.K.); misra@unr.edu (M.M.)

**Keywords:** shot peening, ultrasonic impact peening, laser shock peening, severe plastic deformation, microstructure, fatigue life

## Abstract

Surface modification methods have been applied to metals and alloys to change the surface integrity, obtain superior mechanical properties, and improve service life irrespective of the field of application. In this review paper, current state-of-the-art of peening techniques are demonstrated. More specifically, classical and advanced shot peening (SP), ultrasonic impact peening (UIP), and laser shock peening (LSP) have been discussed. The effect of these techniques on mechanical properties, such as hardness, wear resistance, fatigue life, surface roughness, and corrosion resistance of various metals and alloys, are discussed. This study also reports the comparisons, advantages, challenges, and potential applications of these processes.

## 1. Introduction

In recent years, materials with superior surface integrity have played a paramount role in defense, medical, industrial, and automotive applications because of increased durability and predominant stability during static and dynamic loading. This can be primarily obtained by subjecting the substrate materials to surface modification methods [1]. Tremendous research has been conducted in the past to obtain materials with enhanced surface integrity. Processes, such as deep cold rolling [2], surface mechanical attrition treatment (SMAT) [3], thermochemical methods [4], and severe plastic deformation [5] can potentially alter the surface topography and improve the surface mechanical properties of the substrate. Peening has been widely perceived as a simple, most effective, and industrially reliable surface modification method [6]. Peening techniques can be classified as shot peening (SP), ultrasonic impact peening (UIP), and laser shock peening (LSP).

SP was introduced in the 1950s to enhance the properties of aerospace components. However, this technique has been effectively applied to many engineering materials for the past seven decades. SP treatment changes the surface integrity of target materials by refining the microstructure, developing phase transformations, increasing work hardening, changing surface topography, and inducing residual compressive stress (RCS) [7,8,9,10]. Advanced methods like severe shot peeing (SSP) and micro-shot peening (MSP) processes were introduced to obtain superior surface properties compared to SP [11,12,13,14,15].

Furthermore, peening with high-frequency ultrasonic oscillation came into existence in the early 1960s. It involves a frequency of more than 20 kHz applied using cylindrical working heads to the substrate such that the material is subjected to severe plastic deformation (SPD) and RCS induced on the surface of the substrate. This SPD helps in grain refinement, microstructural modifications, removal of existing tensile residual stress, and closure of existing microcracks [16]. The dense layer formed during UIP on the surface improves hardness, wear resistance, corrosion resistance, and enhances the fatigue life of the substrate [17]. These deformations developed during UIP cause changes in the physical and mechanical properties of the substrate.

To cater to the needs and demands of materials for widespread applications in sectors such as automobile, aerospace, medical, and other industries, LSP came into existence in the late 1960s and early 1970s. This method is prominently used because it can potentially avoid many setbacks of SP and UIP [18]. LSP primarily involves the interaction of high-power density lasers with substrates. In the past, researchers conducted innovative studies using LSP experiments on metals and alloys that are prominently used in marine applications [19,20,21,22], aerospace [23,24,25,26], medical [27,28,29], and industry [30,31,32]. These studies demonstrated the impact of various LSP parameters on mechanical properties and microstructural features.

This review paper aims to provide a comprehensive overview of various peening techniques. Section 2 discusses various peening techniques, such as classical and advanced SP, UIP, and LSP. Each sub-section discusses the fundamental mechanisms associated with the respective peening process and the effect on various mechanical properties, such as hardness, corrosion resistance, fatigue life, tensile strength, residual stress, and wear resistance for different engineering materials. The comparison between classical and advanced SP, UIP, and LSP were demonstrated over a broad spectrum of materials. The advantages and challenges for each of the peening processes were delineated. Recent advances of SP technique, such as warm shot peening (WSP), SSP, and MSP, and LSP techniques, such as laser peening without coating (LPwC), warm laser shock peening (WLSP), cryogenic laser shock peening (CLSP), femtosecond laser shock peening (fS-LSP), laser peen forming (LPF), electro pulsing-assisted laser shock peening (EP-LSP) were elucidated. Finally, applications and future development of peening techniques are explored in Section 3.

## 2. Peening Techniques

Peening is a technique for surface modification that has been widely used over the past five decades. Scholars demonstrated that these peening techniques significantly enhanced surface mechanical properties [33,34,35,36], refined the surface microstructure [37,38,39], induced RCS [40,41,42], and changed the surface topography [43,44]. Generally, SP, UIP, LSP have been the most widely used peening techniques for surface modification. Figure 1 depicts the schematic of SP, UIP, LSP processes. In the SP technique, compressed air accelerates many small hard spherical shots at controlled velocity towards the target material. The interaction between shots and substrate produces dimples on the surface, as shown in Figure 1a. Near the dimple region, a plastically deformed zone, followed by an elastic zone, develops. The recovery process upon rebounding the shots induces RCS on the substrate surface. Peening intensity and peening coverage are the most influential parameters in the SP process. In the UIP process, a peening gun with a tooltip that vibrates at a high frequency from an ultrasonic generator is used for surface modification (Figure 1b). The interaction of vibrating tooltip and substrate surface induces plastic deformation. Significant enhancement in the peening effect can be obtained by amplifying the frequency of vibration with the help of the booster and horn assembly. LSP process takes advantage of the laser-induced shock wave to create RCS and a surface hardening effect on the substrate surface. The laser-matter interaction creates plasma whose expansion is constrained by a tamping material. Due to this, a high-pressure laser-induced shock wave propagates into the target, which causes high strain rate plastic deformation (Figure 1c). Thus, near-surface, RCS, and a work-hardened layer are formed on the target surface. Though peening techniques mentioned above provide grain refinement and RCS through plastic deformation, their effect varies depending upon the chosen material and selected peening parameters. Many controllable parameters are associated with peening, and parameter selection for peening is cumbersome in every peening process [11,12,13]. Inappropriate selection of peening parameters leads to harmful effects, such as reduced service life, increased surface roughness, and decreased component performance [45]. The following section provides a detailed explanation of each peening process, each process’s mechanism, and the effect of these processes on microstructure and mechanical properties on a broad spectrum of engineering materials.

### 2.1. Shot Peening

SP is a predominantly used cold working method to improve the surface mechanical properties of the engineering materials. Spherical shots made of different materials such as metals, glass, and ceramics are used in this process. These shots are directed towards the target material through a gun that is working on compressed air. These shots were impacted on the target material with high velocity so that material is elasto-plastically deformed, and during the recovery process, RCS is induced on the surface [46,47]. The induced stress imparts superior properties and helps to decrease the susceptibility to failure originating from the surface. Kobayashi et al. [48] explained the residual stress mechanism during SP. The SP can effectively hinder the crack propagation and prevent fatigue, fretting, and stress corrosion failure. Many controllable parameters can affect the severity of the peening. These parameters are categorized as the shot, target, and flow parameters. These include peening coverage, peening intensity, peening angle, shot velocity, shot diameter, shot duration, shot material, and target material [49]. Peening intensity and peening coverage are the utmost important parameters in the SP process. An increase in both these parameters can significantly enhance the RCS and RCS layer thickness. However, a contradicting relationship between peening intensity and RCS has also been reported [50]. Earlier researchers conducted trial and error experiments to identify the optimum parameters for peening, which is arduous and time-consuming. Nowadays, scholars demonstrate a finite element model (FEM) and response surface methodology (RSM) for optimization of the peening parameters because of low cost and high efficiency [51,52,53,54]. Single and multi-shot simulations were proposed earlier, however, random multi-shot simulations are more prominent these days. Lin et al. [55] proposed a random multi-shot simulation using Abaqus and correlated the influence of shot velocity and peening coverage on RCS, and dislocation cell size. Their study revealed that an increase in shot velocity leads to finer dislocation cell size, deeper refined microstructure layer, and increased RCS layer thickness. An increase in peening coverage cannot thicken the refined microstructure RCS layer. The notable remark from this study is that a significant increase in RCS in sub-surface is observed with higher peening coverage than higher shot velocity. Wang et al. [56] combined FEM with RSM to optimize the shot parameters for the effective peening 42CrMo specimen, primarily used in river-sea going ships. The optimized shot peening parameters include the shot velocity of 88 m/s, the shot diameter of 0.8 mm, and the coverage ratio of 170%. They reported that with optimized parameters fatigue life of the 42CrMo specimen improved by 104%.

#### 2.1.1. Effect of Shot Peening on Engineering Materials

Researchers conducted SP experiments on various metals and alloys used in industrial, aerospace, medical, and automobile applications. Tadge et al. [33] demonstrated SP experiments on AISI 304 stainless steel (SS) to identify the effect of SP on surface mechanical properties. Their study found that SP caused surface nanocrystalization, which is strengthened by redistribution of carbide particles into the grain from grain boundaries and austenite to martensitic phase transformation. Wang et al. [34] conducted SP experiments on Ti6Al4V and demonstrated the effect of SP on crack propagation during fatigue. Their study disclosed that SP delayed the occurrence of fatigue cracks. Sasikumar et al. [35] showed that SP increased wear resistance on Al7075 hybrid aluminum metal matrix composites. Pfeiffer et al. [36] successfully demonstrated the effect of SP on alumina and silicon nitride ceramics, and they summarized that SP induced RCS about 2 GPa. Kovac et al. [57] conducted SP experiments on AISI 4140 low alloy steel to identify the effect of plastic deformation on corrosion behavior. After SP, the increased corrosion resistance is attributed to the reduction in crystal size, subgrain formation, and surface nanocrystalization. However, some other researchers divulged the deleterious effect of SP, which decreased the corrosion resistance of alloys after SP [58,59].

#### 2.1.2. Advanced Shot Peening

Recently, SSP has been introduced as a method to obtain superior surface property. In this process, intense shot peening parameters are used, which leads to severe surface nanocrystalization and deeper RCS [11,12,13,60,61,62,63]. Chen et al. [64] systematically conducted SSP on Hastelloy X alloy, a nickel-based superalloy, and compared it with SP. Substantial improvement in surface mechanical properties and grain refinement was observed after SSP. They reported an average grain size of 50 nm, RCS at 125 μm deep was 1275 MPa, and the predominant grain refinement mechanism was dislocation movement and deformation twins. Figure 2 indicates transmission electron microscopic (TEM) images from the top surface of SP and SSP treated Hastelloy X. Bright field (BF) image of SP treated Hastelloy X showing dislocation tangles (Figure 2a). Corresponding dark field (DF) image showing lamellar microstructure with high-density dislocations (Figure 2b). The bright field image of SSP treated Hastelloy X is shown in Figure 2c. Selected area electron diffraction (SAED) pattern of the SSP treated sample indicates well-formed rings on the surface. Nanograins were observed on the SSP treated surface and the statistical distribution of grain size inset in (Figure 2d). It is evident from the figure that a significant reduction in grain size was observed after SSP.

Mikova et al. [65] conducted SSP on X70 steel and correlated it with fatigue performance. They stated that, although SSP increased surface roughness, both notched and smooth specimens showed improvement in fatigue strength after SSP. They also recommended that further improvement in fatigue strength can be achieved by higher peening coverage and intensity. Liu et al. [66] observed the effect of SSP on AZ31 (predominantly α-Mg matrix) and AZ91 (α-Mg and β-phases) magnesium alloy. They revealed that SSP on AZ31 caused refinement of α-Mg grains, which enhanced the corrosion resistance. Although SSP refined α-Mg grains in AZ91, no refinement of β- phase was observed, leading to little effect on corrosion resistance. This is because the corrosion resistance of AZ91 greatly depends on size, distribution, and fraction β-phase. Unal et al. [67] described the effect of SSP on microstructure evolution and mechanical behavior of pure Ti. They reported that an ultra-fine grained crystalline structure was observed on the surface after SSP, which has grain size below 10 nm and improved microhardness because of SPD. One of the notable issues associated with SSP is the increase of surface roughness, which is deleterious. The peening effect can be enhanced by impinging the target material with micro-shots—smaller and harder than SP shots—by a method called micro-shot peening (MSP), which reduces surface roughness and enhances surface properties. Harada et al. [14,15] conducted MSP experiments on high-speed tool steel and structural steel. They summarized that significant enhancement in peening is observed with micro-shots.

Li et al. [68] conducted MSP, SP, and their combination technique called dual shot peening (DSP) on EA4T axle steel and correlated it with mechanical properties. They summarized that MSP induced maximum RCS, minimum surface roughness, and high hardness among the three peening methods. DSP’ed specimen showed improved properties compared to SP’ed specimen. The fatigue life of MSP’ed, SP’ed, and DSP’ed specimen were improved by 32, 27, and 24%, respectively, compared to the untreated specimen. SEM observation of fracture surface revealed that fatigue crack originated at the surface and fracture mechanism is identical for MSP, SP, and DSP treated EA4T axle steel. Peral et al. [69] conducted SP and SSP tests on the quenched and tempered 39NiCrMo3 low alloy steel and analyzed the nanocrystalline surface formed by SP and SSP using TEM. They observed that SP treatment favors ferrite amorphization and carbide reduction to the nanometer level. However, during SSP, new carbide nanocrystals were formed from the previously created amorphous matrix. The authors reported significant improvement in hardness, RCS, depth of RCS layer and grain refinement for SSP treated specimen than SP. Liu et al. [70] performed SSP experiments on Mg-8Gd-3Y alloys and discussed microstructural evolution and mechanical properties. The authors explained the formation of nano-size grain refinement on the surface during SSP in three steps. In the initial stage of SSP deformation, twins and dislocations were formed inside large-size grains. Then, the substructure is formed from large grains by the interaction between twin-twin and twin-dislocation cells at a high strain rate. Finally, to accommodate high strain, subgrains began to rotate, and nanograins with high angle grain boundaries were formed. They reported RCS of 205 MPa on the surface, RCS affected to a depth of 250 µm, and 94% improvement in surface hardness was observed. Figure 3 shows the mechanism of grain refinement in Mg-8Gd-3Y.

Bagherifard et al. [71] described the effect of SSP on cast iron and the influence of re-peened severe shot peening (RSSP) on surface roughness. They concluded that SSP caused surface nano crystallization, refined the grain size, caused high work hardening, induced in-depth RCS, and increased surface roughness. The fatigue life of the SSP’ed specimen is deteriorated because of the high kinetic energy of shots during SSP, which led to surface defects and microcracks. However, significant improvement in fatigue life of cast iron specimen is observed after re-peening with ceramic shots on the SSP’ed specimen. Their study recommended RSSP as a way to improve surface roughness and, thereby, fatigue properties. Figure 4 illustrates the scanning electron microscopy (SEM) observation of different peening processes on cast iron. It is evident that RSSP specimens have better surface integrity compared to SSP specimens. Gao et al. [23] conducted fatigue experiments on machined, shot-peened, and laser peened 7050–T7451 aluminum alloy plates. 

They summarized that both peening methods improved the fatigue performance of the specimen with dominated fatigue properties obtained for laser peened substrate. Laser peened and shot-peened substrate showed 42 and 35% higher fatigue strength than machined samples. They also reported that ceramic and double shot-peened specimen have similar fatigue properties compared to laser peened substrates. Table 1 shows the influence of classical and advanced SP techniques on surface mechanical properties and microstructural features. Han et al. [72] SP experiments on austempered AISI 5160 steel revealed that SP treatment introduced observable plastic deformation and improved surface hardness of soft austempered specimens. They reported a wear reduction of 73% on shot-peened austempered specimens compared to un-peened austempered specimens.

SP is a simple, flexible, cost-effective, robust, and highly efficient peening method, especially for fatigue enhancement of metals [55,73]. Even though shot peening provides these many advantages, there are some limitations. The peening intensity determined by almen gauge does not guarantee uniform peening intensity, depth of induced compressive stress is limited, and it leads to an increase in surface roughness and requirement of secondary treatment for wear applications [74,75].

### 2.2. Ultrasonic Impact Peening

In this cold working method, a carburized steel tool is used to induce ultrasonic impacts on the target material. This method finds wide applications in aerospace, automotive, marine, and civil structures [76,77,78,79,80]. The primary components in ultrasonic peening equipment involve a high voltage power supply to operate the peening gun, a transducer, a concentrator, and an impact tool. Figure 5 illustrates the ultrasonic peening equipment.

Statnikov et al. [81] explained the physics and mechanism of ultrasonic impact. When the electric power supply is turned on, a standing wave is produced by the transducer, which is further amplified by the concentrator, and then it is transferred to the tool. The tool transmits this vibration as an impact to the surface, and thus the material is subjected to SPD. There are two types of transducer which were commonly used in the UIP process. A magnetostrictive transducer converts the electric energy to high amplitude high-frequency vibrations at the tip of the tool. It is of utmost importance to have a cooling system coupled with the magnetostrictive transducer to avoid excess heat and minimize the energy loss in the transducer. Since the magnetostrictive transducer needs to have a cooling system, it is more bulky and costly. Nowadays, piezoelectric transducers are widely adopted because they are lighter and easy to handle. The promising advantages of UIP are process parameters that can be easily controlled, minimum energy consumption, wide industrial applications, and potentially no pollution [37,38,39].

#### 2.2.1. Effect of Ultrasonic Impact Peening on Engineering Materials

Scholars have demonstrated the usage of UIP to improve various mechanical properties of engineering components [82,83,84]. Chen et al. [37] studied the effect of UIP duration and peening distance on pure copper and correlated its effect with microstructure and mechanical properties. An increase in both parameters causes an increase in thickness of the fine layer on the surface of copper, which ultimately enhanced hardness by 233% and tensile strength by 17%. Wu et al. [38] performed UIP on aluminum alloy 7075 and reported that UIP caused ultra-fine grain microstructures to a depth of 62 µm. Furthermore, severe straining by UIP resulted in the formation of microbands, dislocation tangles, dislocation cells, highly misoriented boundaries, and equiaxed sub-grains. The second phase trapezoidal particles present in the aluminum matrix act as an emission source of dislocation. Using electron diffraction patterns, these trapezoidal particles were identified as Al_2_Cu. Dislocation cells (DC) and dislocation tangles (DT) are responsible for work hardening during straining and they are formed interior of the aluminum matrix grains. Figure 6a shows the TEM emission source of dislocation, and Figure 6b shows DC and DT in the aluminum matrix.

Abdullah et al. [39] studied the influence of UIP on welded stainless steel 304 sheets. The results indicate that UIP increased the fatigue life of weld joints by 120%, fatigue strength by 29%, hardness in weld metal and weld toe were higher than base metal, improved corrosion resistance. Microstructural modification developed due to UIP, including nano grain size, strain induced martensite, and deformations twin formation which is the potential reason for superior mechanical properties. Figure 7a shows the SEM images of the weld toe of SS 304 sheets. It is evident that surface microcracks of 21 µm are present in the weld toe of SS 304. These cracks can lead to catastrophic failure of the component during service. Figure 7b shows SEM of UIP treated welded SS 304 sheets. The authors reported that UIP treatment caused the closure of surface microcracks and modified weld toe curvature of welded SS 304 sheets.

Ling et al. [40] carried out UIP experiments on TIG welded 304 SS sheets and elucidated that UIP induced a hardened layer on the surface and ultra-fine grains observed in the hardened layer, which increased the corrosion resistance. Wang et al. [82] conducted UIP and LSP experiments on AISI 316 stainless steel and compared microstructure and mechanical properties. The observed results indicate that the dominant mechanism of plastic deformation in the case of UIP processed substrate is dislocation whereas, twinning for LSP. The magnitude of induced compressive stress, yield strength, and hardness were higher for LSP treated samples because of grain refinement and grain boundary strengthening. Li et al. [83] conducted UIP on 301 SS and observed the effect of UIP on magnetic properties and microstructural characterization. They reported that saturation magnetization, coercivity, grain size, and martensitic ratio in the transition layer were significantly changed after UIP. The magnetization variation depends on both microstructures as well as grain size. Figure 8a demonstrates the relationship between magnetization and martensitic ratio, while Figure 8b shows the relationship between coercivity and grain size for two different peening times. They observed an increase of 144% in hardness and hardening depth of 450 µm after UIP. Zhu et al. [84] conducted UIP experiment on pure Ti and demonstrated the effect of peening distance, peening duration, and shot diameter on surface hardness and microstructure. Their study concluded that with an increase of peening duration and shot diameter, surface hardness increases, whereas it decreases with the increase of peening distance.

They also predicted a model based on experimental results to determine the surface hardness under different process parameters. Tian et al. [85] provided insight into the application of UIP on cold metal transfer (CMT) welded 6061 aluminum alloy, and they concluded that UIP can effectively eliminate the pores in the weld bead close to the weld surface, and changed the morphology of pores in the center of the weld bead. After UIP, the elastic modulus of weld bead and wear resistance of weld were enhanced because of the reduction in porosity content. Improvement in hardness can be attributed to the work hardening and reduction in grain size developed during UIP. Figure 9a,b show the SEM of pores in three different regions of weld bead before and after UIP treatment. It is evident that after UIP treatment, number of pores reduced in weld bead close to weld surface.

Kumar et al. [86] reported improvement in corrosion resistance after UIP on Ti6Al4V alloy. Moreover, UIP technologies are widely used as a method to decrease stress concentration in weld toe and as a post welding treatment to enhance the strength and hardness of weld joints [39]. Table 2 represents the effect UIP process on mechanical and microstructural features on various engineering materials [87,88,89,90,91].

Liu et al. [92] conducted UIP experiments with Ti6Al4V pin on EQ70 high strength low alloy steel (HSLA) to generate a cladding layer on the surface. The authors reported the presence of mixed oxides, such as TiO, TiO_2_, FeO, and Fe_2_O_3,_ on the hard-cladding layer. The authors revealed that mechanochemical oxidation is the main mechanism for the formation of the cladding layer. An increase in ultrasonic impact intensity increases the thickness of the cladding layer and it follows a linear relationship. Figure 10 represents the polarization curves for cladded EQ70 HSLA steel obtained by electrochemical corrosion test. Lower the corrosion current density implies higher corrosion resistance. From the figure, it is clear that corrosion current density is decreased with an increase in impact intensity. When the impact intensity is over 1.33 s/mm^2^, the cladding layer consists of many pores and cracks due to SPD. The authors recommended that peening intensity in the range 1 to 1.33 s/mm^2^ can provide the best corrosion-resistant properties due to the compact cladding layer.

Li et al. [93] conducted UIP on AlxCoCrFeMnNi (x = 0, 0.5, 1.0 and 1.5) high entropy alloys (HEA) and investigated the microstructure and mechanical properties. The authors reported that UIP did not affect the phase compositions. However, long-strip structures in the hardened layer instead of short rod structures were observed. The microhardness improved by 92% due to precipitation strengthening and grain refinement. Significant reduction in surface roughness and high corrosion resistance was observed on UIP treated specimen compared to as-cast alloys. Zhang et al. [94] conducted UIP on laser cladded AlCoCrCuFeNi HEA and correlated with mechanical properties. They observed a significant enhancement in hardness, 47% grain refinement, a hardened layer of 15 μm, higher corrosion resistance, and lower surface roughness. Figure 11a shows SEM of laser cladded AlCoCrCuFeNi HEA. It is evident that dendrite structures and some thick strip-like structures can be observed. Figure 11b shows SEM of UIP on laser cladded AlCoCrCuFeNi. It is obvious that due to plastic deformation, gradient microstructures were formed on the surface. After UIP treatment, the strip structure is converted to the short rod structure and dendritic structure.

### 2.3. Laser Shock Peening

LSP is a prominently used peening method to induce high magnitude RCS deeper and more uniformly into the surface of the substrate. LSP fundamentally involves the interaction of laser and sacrificially coated target material in a confined medium. The sacrificial coating helps to prevent thermal effects like laser ablation, melting, and generation of tensile stress, and enhances the peak pressure of the shock wave induced during direct laser interaction [95]. Materials like aluminum, copper, zinc, and black paint are effectively used as a sacrificial coating on target materials [74,75,96,97,98]. The black coating is considered an ideal coating material because of its 100% ability to absorb laser energy, which is experimentally proven by Hong et al. [99]. When the sacrificially coated target material is irradiated with a laser beam in a confined medium, laser and coated material interaction cause vaporization of coated material, producing laser-induced plasma. Sundar et al. [74] explained the two methods and mechanisms of plasma creation during LSP.

The presence of the confined medium prevents the free expansion of the plasma, leading to the formation of the high-pressure shock wave. Researchers also provided deep insights into the various confining media such as water, a variety of glasses, and silicon rubber [100,101,102,103]. The restricted expansion of plasma in the confined medium helps to enhance the peak pressure and pulse duration of the shock wave. The high-pressure shock wave eventually hit on the target material and subject to SPD near to the surface with a strain rate in the order of 10^6^/s to 10^7^/s [104].

To cause plastic deformation and thereby microstructural modification, it should be of utmost importance that the pressure of the shock wave exceeds the dynamic yield strength of the target material [74]. The residual stress induced on the substrate due to LSP treatment is likely highly related to the peak pressure of the shock wave.

#### 2.3.1. Effect of Laser Shock Peening on Engineering Materials

Many researchers have experimentally proved that laser-assisted peening process is superior in obtaining properties like increased wear resistance [21,24,26,31], improved hardness [21,24,25,26,27,28,30], increased fatigue life [23,26], enhanced corrosion resistance [20,22,32], increased yield strength [23,27,30], improved surface roughness [20,24,26,31], and refined microstructure [25,27,28] in metals and alloys. Researchers demonstrated many materials that are successfully peened using LSP techniques including, but not limited to aluminum alloys [21,23,24], different steel grades [20,22,27,32], titanium alloys [16,17,25], magnesium alloys [28], ceramics [29], super alloys [30], and brass [31]. Recently, LSP techniques have been widely applied as a method for post-processing of welds [105] and on additively manufactured (AM) components [106]. LSP techniques successfully imparted significant enhancement in surface mechanical properties of weld joints of Laser beam welding (LBW) [107,108,109,110], Tungsten inert gas welding (TIG) [111,112,113], and Friction stir welding (FSW) [114] because of the superiority of LSP compared to other methods of peening. The inherent advantage of LSP compared to SP and UIP are the introduction of compressive stress to in-depth, the magnitude of compressive stress, better surface finish, less damage to the initial surface, accuracy, and flexibility [18,104]. Table 3 illustrates the different process parameters used by researchers to conduct LSP experiments on various materials and their findings.

Luo et al. [27] explained the effect of LSP on the surface mechanical properties and microstructure on the ANSI 304 austenitic SS. It is observed that nano hardness and elastic moduli of single-shot LSP processed material are higher than the unprocessed material. Figure 12a shows the relationship between hardness and elastic moduli with applied load.

The microstructural features indicate that mechanical twins (MT) were observed with nanometer sized twin spacing. As the twin spacing decreases, the hardness increases. LSP also refined the microstructure and fraction porosity which led to an increase in elastic moduli. Figure 12b illustrates the RCS with and without LSP. It is evident that the induced RCS during LSP is up to a depth of 900 µm, and the maximum residual stress is 305 MPa. Peyre et al. [115] demonstrated the influence of SP and LSP on 316L steel. Microstructural characterization of LSP treated substrate revealed deformation twins and persistent slip bands. SP treated samples contain a dense array of slip bands shown in Figure 13a whereas strain-induced martensitic needles formed during LSP are shown in Figure 13b. SP generates larger plastic deformation. Increased pitting potential was observed for LSP treated specimen compared to SP. 

Ruschau et al. [116] reported the enhancement of fatigue life of notched Ti6Al4V alloy during LSP. They concluded that the LSP processed notched alloy shows greater resistance to fatigue crack propagation compared to the substrate. This is due to the higher magnitude of RCS developed by LSP, which suppressed the local stress during notching. Figure 14a,b show TEM of LSP processed and unprocessed Ti6Al4V alloy. It is evident that LSP caused a significant increase in dislocation density, and it affected microstructure at the subgrain level.

Hatamleh [96] demonstrated that LSP can prominently reduce the fatigue crack growth rate, thus improving the fatigue life of friction stir welded AA 2195 joints compared to SP. LSP predominantly reduced the tensile stress generated during welding, and it is observed that LSP can significantly induce higher and deeper RCS compared to SP. Ganesh et al. [117] conducted SP and LSP experiments on SAE 9260 spring steel extensively used in the automotive industry. The observed results indicate that LSP increased fatigue life and enhanced fatigue performance due to the better surface finish without induced defects on the surface during LSP compared to SP. Surface roughness values almost doubled for SP processed substrate, whereas surface roughness value during LSP is similar to the substrate. The surface roughness profiles of base material (BM), SP and LSP treated specimen is shown in Figure 15.

Hackel et al. [118] reviewed studies on the influence of SP and LSP for post-processing of AM components. Their study divulged that both SP and LSP played a pivotal role in increasing the fatigue life and fatigue performance of AM 316L SS. LSP in particular is a potential tool to prevent premature fatigue failure due to the stress concentration from fillets and notches in the AM component. They also pointed out that LSP can be implemented as a potential tool for precise shape correction in AM components. Although LSP can improve the fatigue life of components, fatigue life enhancement by LSP on materials working at high temperatures is limited [119]. This is because due to high temperature, the materials relieve the RCS and lose stability of microstructure.

Lu et al. [120] studied the influence of LSP on the hot corrosion behavior of selective laser melting (SLM) of Ti6Al4V titanium alloy. The SLM’ed Ti6Al4V alloy predominantly consists of lamellar and acicular martensite structures. After LSP treatment, a large number of dislocations and nano-twins were observed, which improved the grain boundary effect on the surface layer. The authors reported that LSP significantly enhanced hot corrosion behavior by the combined plastic deformation and grain boundary effects. Figure 16a,b show the TEM of the LSP’ed SLM component specimen. A uniform microstructure with a large number of needle-shaped mechanical twins (MT), high-density dislocations, and refined acicular martensite after LSP treatment were observed, which contributed to grain refinement and superior mechanical properties.

Zheng et al. [121] investigated the effect of LSP on high-temperature (200, 300, 400, and 500 °C) tensile properties on new-generation lightweight 2060 Al-Li alloy, which is predominantly used in aerospace and military applications. Significant improvement in tensile properties was observed for LSP treated tensile tested samples below 400 °C. The authors summarized that dislocation strengthening is the primary reason for superior high temperature tensile properties. Li et al. [122] conducted LSP on aluminized AISI 321 stainless steel and studied the high cycle fatigue behavior. The authors reported about a 200–230% improvement in fatigue life, which is attributed to the formation of deformation twins and high RCS at the SPD layer. Lu et al. [123] implanted diamond nanoparticles using LSP on the surface of 20Cr2Ni4A alloy steel and studied the mechanical and wear properties. They revealed that the crater depth induced by LSP during nanoparticle implantation reduced to 27.3 µm due to the buffering effect of nanoparticles compared to LSP without nanoparticle implantation. The nanohardness and elastic modulus were enhanced by 31.6 and 13.6% compared to the untreated sample. The nanoparticle implantation using LSP caused a reduction in surface roughness, wear loss was reduced by 11%, and the wear mechanism changed to a mix of abrasive and adhesive wear. Figure 17 shows the schematic of diamond nanoparticle implantation using the LSP technique. When a high-intensity laser beam penetrates the water confinement layer and strikes the aluminum foil, the aluminum foil breaks away. The high-pressure plasma formed acts on the dispersed diamond nanoparticle on the substrate surface and are deposited on the substrate.

Dhakal et al. [105] reviewed the use of LSP in various weld joints and summarized that it can provide quality weld joints with improved mechanical properties in the weldment. Table 4 demonstrates the benefits of LSP as a post-weld treatment method and the corresponding enhancement of various properties of weld joints in different metals and alloys.

#### 2.3.2. Recent Developments in LSP

In laser processing, applying a sacrificial coating to intricate and complex shaped geometries is difficult, which reduces the efficiency of LSP. Mawaad et al. [124] proposed a new technique of LPwC. They revealed that LPwC can induce in-depth and stable RCS onto the surface of the substrate compared to SP and UIP. This LPwC technique can successfully be applied to welds, and it can enhance the fatigue life of welded components [125,126]. Researchers demonstrated the use of square-shaped laser beams instead of circular laser beams. The square-shaped laser technique showed improved coverage, significant overlap, and enhanced surface quality [75,127] Altenberger et al. [128] revealed the residual stress relaxation in LSP treated 304 SS and Ti64 alloy operated at a temperature range of 550−600 °C. To revive the fatigue life, a new method of peening called WLSP and thermal engineering LSP were introduced [129,130]. The WLSP combines the advantages of LSP, DSA, and dynamic precipitation (DP). Ye et al. [131] successfully demonstrated fatigue life enhancement of AISI 4140 steel during WLSP. They concluded WLSP treated substrate has better fatigue properties than LSP substrate. The DSA helped to increase the dislocation density and stabilized the dislocation structure by pinning mobile dislocations. Moreover, ultra-fine precipitates aid in stabilizing the microstructure and retaining the residual stress. These studies indicate that WLSP plays a crucial role in high-temperature fatigue compared to LSP. Figure 18a demonstrates the residual stress relaxation at an annealing temperature of 300 °C and Figure 18b indicates the higher magnitude of residual stress retained by WLSP substrate compared to LSP during cyclic loading. WSLP can play a prominent role in high-temperature cyclic loading applications. Meng et al. [132] demonstrated that WLSP treated Ti6Al4V at 350 °C showed improved vibration fatigue performance because of the grain boundary migration and high strain rate in the β phase which ultimately increased the volume fraction of α phase. Appropriate selection of temperature for WLSP is an important consideration because it can affect the depth, magnitude of RCS, and microstructure. Similar to WLSP, LSP can be performed at cryogenic temperature in the liquid nitrogen atmosphere called CLSP. Li et al. [133] conducted CLSP experiments on TC6 alloy and explained that CLSP treatment leads to an increase in dislocation density, deformation twins, and surface RCS, improving high cycle bending fatigue strength.

One of the recent applications of LSP is to increase the fatigue life of SLM’ed components by a method called 3D LSP technique [134]. SLM is a mature method of AM, and it can create more complex and intricate geometries. One of the problems faced by SLM components is tensile residual stress and porosity content in the development phase. This residual stress can create many problems in SLM manufactured components, such as delamination and process failure [135]. Lu et al. [136] explained that integration of SLM and LSP techniques can remove the harmful residual tensile stress and can induce beneficial RCS, hence, the fatigue life of the component can be enhanced. Figure 19a shows an integrated system of SLM and LSP, and Figure 19b indicates the scanning direction during the SLM method. Kalentics et al. [134] described that 3D LSP is a hybrid method of AM in which periodic shocks are applied to the selective laser melting (SLM) method. Their experiment with 3D LSP treatment on 316 L SS specimens showed remarkable improvement in mechanical properties. They revealed that bending fatigue life increased by 15 times and showed a 44% increase in fatigue limit in unmachined samples—these results are fascinating in the case of machined samples. Moreover, it increased crack initiation time and reduced crack propagation rate compared to LSP. Kalentics et al. [137] conducted 3D LSP experiments on a nickel-based superalloy CM247LC, which is prone to cracking during fusion welding and SLM.

They summarized that the 3D LSP technique effectively healed the cracks by 95%. Nakano et al. [138,139] introduced the concept of fs-LSP on SS without coating, and they revealed that fs-LSP is a potential technique that can enhance the surface hardness at lower laser pulse energy. Li et al. [140] demonstrated the effect of confining medium and sacrificial coating on 304 SS with fs-LSP. Figure 20a,b represent the surface hardness and surface roughness of fs-LSP processed SS 304 at different processing conditions. The fs-LSP treatment on 304 SS without confining medium and protective coating showed a 45% improvement in hardness. When water is used as the confining medium, the shock wave propagation distance is less than the thickness of the confining medium and due to water ionization, about 98% laser energy gets absorbed which leads to a poor peening effect. Although the surface roughness increased when peening is conducted in the air without coating, this is acceptable for many applications because only a small area was affected due to the laser. Li et al. [140] recommend fs-LSP as ideal to operate in air without any sacrificial coating. Moreover, a laser-based method of forming called LPF is adopted for accurately bending, shaping, and forming sheet materials using high power density lasers [141]. In this method, the substrate is covered with a sacrificial coating and in a confining medium with both or one end clamped. LPF does not produce any thermal effect, rather, it is a purely mechanical process in which laser-induced shock wave produces the bending.

Among LPF, femtosecond laser peen forming (fs-LPF) and heat-assisted nanosecond laser peen forming (ns-LPF) techniques are prominently used [142,143]. Hu et al. [144] explained stress gradient and shock bending as two mechanisms of bending during LPF. The introduction of high residual stress can easily modify the curvature during formation with high power density lasers. The conventional forming techniques can induce residual tensile stress on the surface and lead to stress corrosion cracking (SCC). LPF has widespread industrial applications in the field of automobiles, microelectronics, shipbuilding, and aerospace. LST is a prominent method used to enhance the tribological performance of engineering materials. Presently, three methods of LST are predominantly used, such as LST by direct laser ablation, laser interference, and laser shock processing. Mao et al. [145] developed a novel method called indirect laser shock surface patterning (LSSP), which can combine the benefits of patterning and surface strengthening simultaneously. The mechanism of indirect-LSSP techniques is explained [146]. Figure 21 illustrates the schematic of the indirect-LSSP process. 

They reported enhanced wear resistance on AISI 1045 sheets of steel after the indirect-LSSP technique because of the formation of anti-skew surfaces with arrays of micro-indentations. Another novel scalable LST process called direct laser shock patterning (LSSP) integrates the strengthening and patterning during laser shock processing. The mechanism of direct-LSSP is elucidated [147]. Zhang et al. [147] demonstrated the effect of direct-LSSP on AZ31B magnesium alloy. They summarized that due to direct LSSP, a skew surface of patterned micro protrusion formed on the surface, which enhanced the surface hardness. Figure 22 indicates the optical micrograph of the substrate and direct-LSSP processed substrate at different intensities.

An increase in laser intensity increases the twins in the bump, which enhanced the surface hardness. EP-LSP is an innovative method to enhance the plasticity of metals [148]. In low plasticity materials, it is difficult to induce RCS through LSP. In this method, the substrate is subjected to resistive heating by the simultaneous application of high-frequency short-pulsed current with high strain rate plastic deformation. They concluded that EP-LSP is more beneficial compared to CC-LSP in reducing the flow stress of materials and thus improving the plasticity. Moreover, LSP techniques are used to induce plastic deformation in ceramics. Wang et al. [149] LSP experiments on polycrystalline *α*-SiC ceramics showed increased dislocation density near-surface, grain boundaries, and presence of stacking fault surrounded by partial dislocations as direct evidence of localized plasticity due to LSP. Figure 23 shows the TEM images in polycrystalline *α*-SiC ceramics after LSP treatment. It is evident that there are multiple dislocations near the surface. The fracture toughness and bending strength of the *α*-SiC ceramics after LSP were improved by 67% and 17%. Table 5 represents the advances in LSP techniques.

Even though LSP offers a wide variety of benefits over classical SP, advanced SP, and UIP, some potential challenges are associated with LSP techniques. Dimension variation of the component during LSP is one of the critical issues. Researchers revealed the dimensional variation in the leading edge of an airfoil due to compressive stress in the radial direction [150]. Moreover, when thin sections are peened, there is a chance that tensile stress resides beneath the peened surface. This usually occurs when the RCS does not distribute equally across the section thickness. This situation can be either prevented by implementing LSP on both sides [151] or by constraining the tensile stresses. The over-processing of LSP could produce an internal rupture of the substrate [152]. However, this can be tackled by careful selection of process parameters. The cost associated with LSP techniques is higher compared to other peening methods.

## 3. Applications and Future Directions

SP is widely used in automobile industries on parts, such as axle, clutch, coil springs, shafts, chassis, engine housings, cams, transmission gears, cylinder heads, piston, connecting rods, and wheels. SP can enhance the surface mechanical properties and helps to withstand static and dynamic loading during service. Wu et al. [47] conducted SP experiments on carburized 18CrNiMo7-6 steel, which is commonly used in power transmission elements, such as gears, bushings, and bearings. They summarized that after SP, improvement in hardness and RCS were observed. In aerospace applications, SP techniques are applied to landing gears, engine parts, aircraft wing skins, rotor blades of the compressor, and turbines. Nie et al. [153] studied the high cycle fatigue behavior of SP treated 3Cr13 high strength spring steels, which are widely used in spring components in aerospace applications. They revealed that fatigue specimens failed at low-stress amplitude. Wick et al. [154] elucidated that WSP on AISI 4140 steel can have better fatigue properties due to DSA, which stabilizes the residual stress compared to SP. Peral et al. [155] conducted WSP experiments on AZ31B magnesium alloy and revealed an increase in hardness at higher WSP temperature. Harada et al. [14,15] conducted WSP with micro-shots on high-speed tool steel and structural steel. They concluded a significant improvement in peening effect and enhancement in surface mechanical properties.

UIP technique is used in various fields of application such as automobile, aerospace, biomedical chemical, and manufacturing industries. Presently, AM components are receiving increased attention due to flexibility and ease of manufacturing. Porosity is a major threat faced by the AM components. Recently, UIP techniques have been widely used as a method to reduce porosity and enhance mechanical properties. Tian et al. [156] conducted UIP experiments on wire arc additively manufactured (WAAM) aluminum alloy. They revealed that after UIP, periodic distribution of refined equiaxed, deformed, and dendritic grains was observed. The porosity developed during WAAM was significantly reduced after UIP treatment. Zhang et al. [157] conducted experiments to identify the effect of UIP on microstructural evolution and corrosion resistance of Ti6Al4V alloy—predominantly used in the aerospace application and produced by SLM—and showed that UIP changed the surface roughness, surface hardness increased by 25%, introduced a significant amount of RCS, and decreased the corrosion current density, which subsequently enhanced corrosion resistance. UIP is used as a method to remove the residual tensile stress on weld joints and thereby enhance the joint efficiency and performance of welded structures. Abdullah et al. [39] studied the influence of UIP on welded SS 304 sheets. The results indicate that UIP increased the fatigue life of weld joints by 120%, fatigue strength by 29%, increased hardness in the weld metal, weld toe, and improved corrosion resistance. Lago et al. [158] reported improvement in fatigue life of high strength steel weldments after UIP.

LSP techniques have wide application in aerospace, nuclear, biomedical, marine, and automotive industries. In the aerospace industry, LSP is used in passenger and military flights to enhance the fatigue performance of the compressor blade, landing gears, shafts, valves, discs, and complex forming of wing surfaces [159,160]. Hammersley et al. [159] demonstrated the effect of LSP on fatigue strength of turbine fan blades, which are made of Ti6Al4V. They summarized that after LSP, enhancement in fatigue strength was two times higher compared to SP. Researchers also showed that LSP can successfully be applied to thin wall welds of Ti6Al4V, which are widely used in the components of aero engines. Shi et al. [112] revealed enhanced fatigue strength, surface hardness, and grain refinement, after LSP treatment on thin wall TIG-welded Ti6Al4V. LSP is widely used in post-processing of welds in different industries [105]. In nuclear industries, stress corrosion cracking (SCC) is a major threat to nuclear reactors and canisters. Wei et al. [161] demonstrated that after LSP treatment on AISI 304 SS, a decrease in SCC susceptibility was shown when tested in acid chloride. Lu et al. [162] showed that LSP treatment on U bend samples of ANSI 304 SS enhanced the resistance to SCC. In the health sector, biodegradable orthopedic implants are subjected to LSP to enhance corrosion resistance and fatigue performance. Sealy et al. [16] conducted LSP experiments on novel biodegradable magnesium-calcium alloy implants. They concluded that LSP enhanced tribological, fatigue, and corrosion properties of implants, hence revision surgeries can be avoided. Wu et al. [163] reported that LSP experiments on Ti-based orthopedic implants produced nano architectures, which enhanced biocompatibility and corrosion resistance. Keshavarz et al. [164] conducted ultra-short pulsed laser irradiation experiments on Si-based bio-template to modulate the interaction between cells and bio template for biological applications. They revealed that the laser interaction caused residual stress onto the bio template, recrystallization of Si, and enhanced directional cell alignment. They pointed out that cell compatibility can be expressed as a function of cell migration and cell alignment, which is directly related to induced residual stress during laser irradiation. They summarized that by varying the laser parameters, cell migration could be controlled.

In the future, the selection of process parameters for peening complex and intricated shapes will be based on machine learning and artificial intelligence coupled with high performance computing. In this way, a significant amount of time and cost can be saved. Moreover, combination techniques like peening and other surface modification techniques will become more prominent to enhance the surface mechanical properties. Amanov et al. [165] conducted a combination of SP and ultrasonic nanocrystal surface modification (UNSM) on AISI 304 to analyze the fatigue performance. They summarized that the combination of SP and UNSM enhanced the fatigue strength compared to SP. Asquith et al. [69] studied the combined effect of SP and Plasma Electrolytic Oxidation (PEO) on corrosion resistance of 2024 aluminum alloy. They reported that duplex treatment increased the corrosion resistance compared to PEO. Tsuji et al. [166] demonstrated the enhancement in wear and fatigue properties on Ti6Al4V subjected to plasma-carburizing (PC) and SP. Even though some of the recent innovations in peening techniques are realized in labs, in the future, full-scale implementation will develop, and industries can utilize these techniques to cater to their needs and demands.

## 4. Conclusions

In this review paper, a widely perceived surface modification technique called peening was explained in detail. Surface modification methods play a paramount role in enhancing the surface mechanical properties of the engineering materials used in various fields of applications. The three major peening techniques, namely classical and advanced SP, UIP, and LSP, have been discussed. The fundamental mechanism of each peening process was elucidated. The effect of peening on a wide spectrum of engineering materials, including different steel grades, titanium alloys, magnesium alloys, aluminum alloys, ceramics, superalloys, and SLM components, was successfully demonstrated. The effect of these peening processes on surface mechanical properties and microstructural features was explained. Some potential applications, advantages, and challenges of peening techniques were incorporated. Recent advances of SP technique, such as WSP, SSP, and MSP were included. Advanced LSP methods like LPwC, WLSP, CLSP, FS-LSP, LPF, EP-LSP, LST by direct-LSSP, and indirect-LSSP were explained in detail. In the future, the process parameter selection for peening solely depends on machine learning and artificial intelligence. Some of the recent innovations in the peening methods have been realized in labs, but these innovations will be available in full-scale industrial purpose in the future. To obtain superior surface properties, a combination of peening and other surface modification methods will come into practice. This review article can offer better insights when choosing the peening techniques for various industrial applications.

## Figures and Tables

**Figure 1 materials-14-03841-f001:**
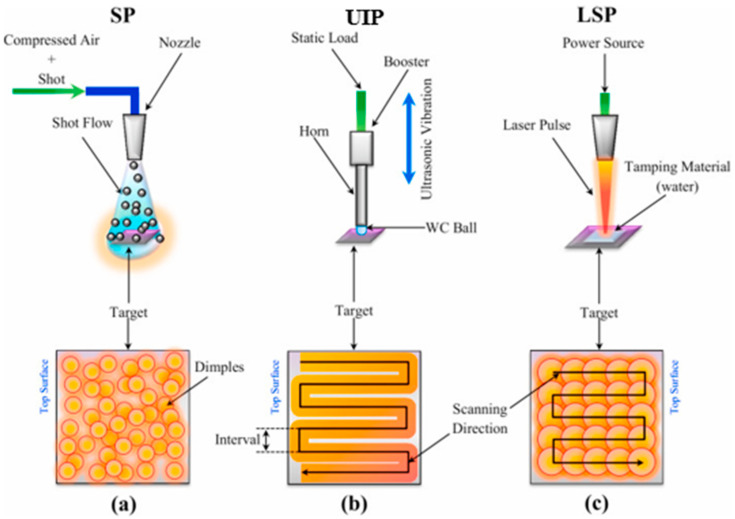
Schematic illustration of different peening techniques and the corresponding plastically deformed top surface of the target material: (**a**) SP; (**b**) UIP; (**c**) LSP. Reproduced with permission from [44]. Copyright Elsevier, 2021.

**Figure 2 materials-14-03841-f002:**
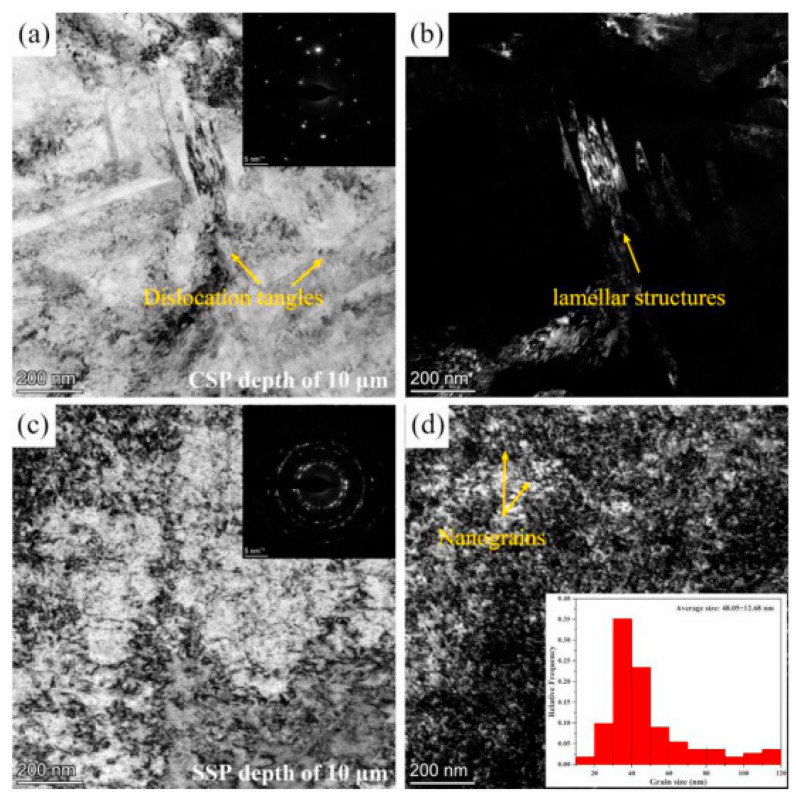
TEM bright field (BF) and the corresponding dark filed (DF) image from the topmost surface of treated Hastelloy X alloy: (**a**,**b**) SP; (**c**,**d**) SSP sample. Inset in Figure 2d is the statistical distribution of the grain size. Reproduced with permission from [64]. Copyright Elsevier, 2021.

**Figure 3 materials-14-03841-f003:**
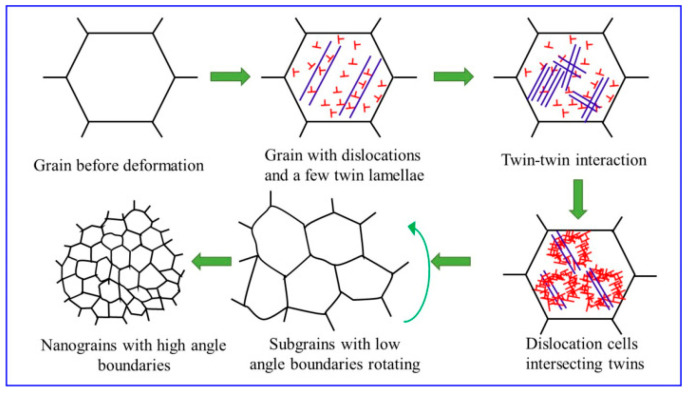
The illustration of grain refinement mechanism of Mg-8Gd-3Y alloy during SSP. Reproduced with permission from [70]. Copyright Elsevier, 2020.

**Figure 4 materials-14-03841-f004:**
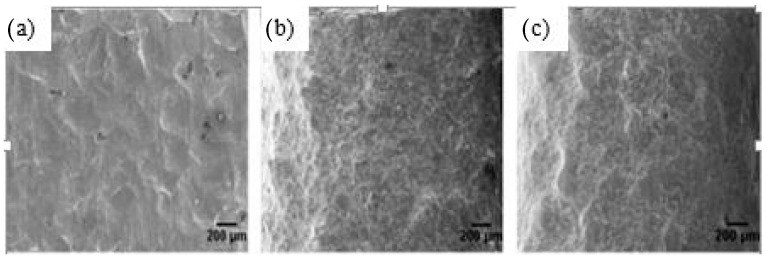
SEM of cast iron surfaces developed by (**a**) SP; (**b**) SSP and (**c**) RSSP. Reproduced with permission from [71]. Copyright Elsevier, 2014.

**Figure 5 materials-14-03841-f005:**
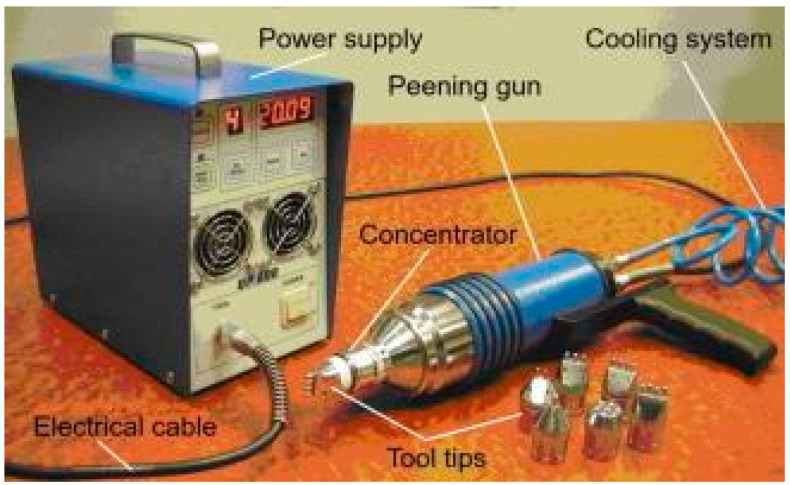
Ultrasonic peening equipment. Reproduced with permission from [80]. Copyright Elsevier, 2015.

**Figure 6 materials-14-03841-f006:**
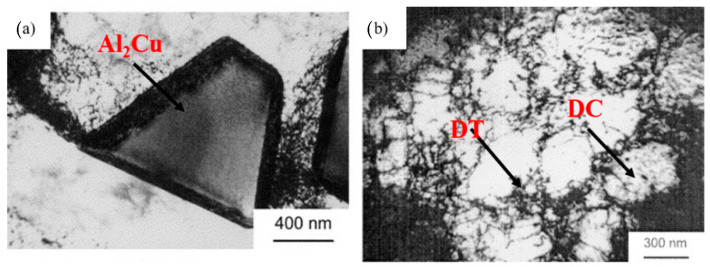
TEM showing (**a**) trapezoidal Al_2_Cu particle as emission source of dislocations (**b**) DC and DT in the aluminum matrix. Reproduced with permission from [38]. Copyright Elsevier, 2002.

**Figure 7 materials-14-03841-f007:**
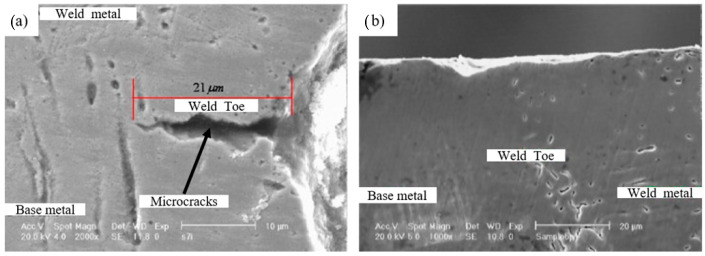
SEM of SS 304 sheets (**a**) microcracks in weld toe; (**b**) after UIP treatment closure of microcracks and modified weld toe curvature. Reproduced with permission from [39]. Copyright Elsevier, 2012.

**Figure 8 materials-14-03841-f008:**
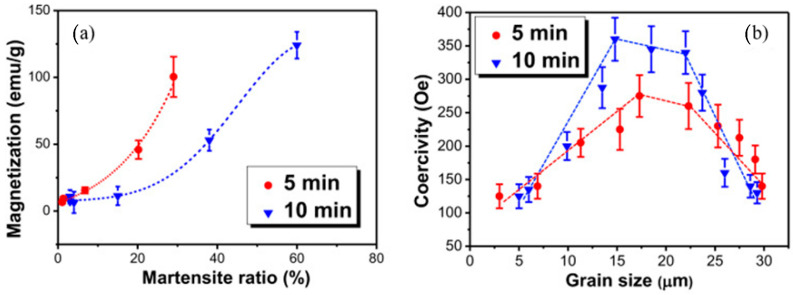
Comparison of the data at the same depth for (**a**) the variation of magnetization with martensite ratio; and (**b**) variation of the coercivity with grain size. Reproduced with permission from [83]. Copyright Elsevier, 2020.

**Figure 9 materials-14-03841-f009:**
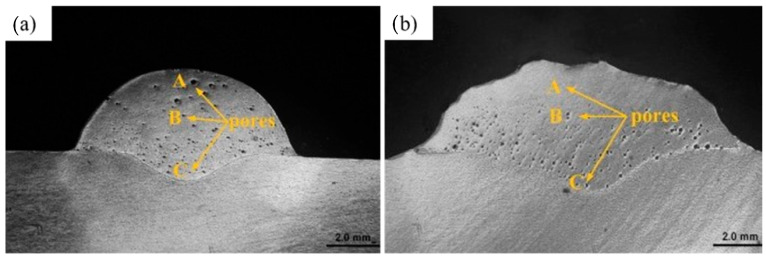
SEM of weld bead of CMT welded 6062 aluminum alloy (**a**) without UIP; (**b**) with UIP. Reproduced with permission from [85]. Copyright Elsevier, 2018.

**Figure 10 materials-14-03841-f010:**
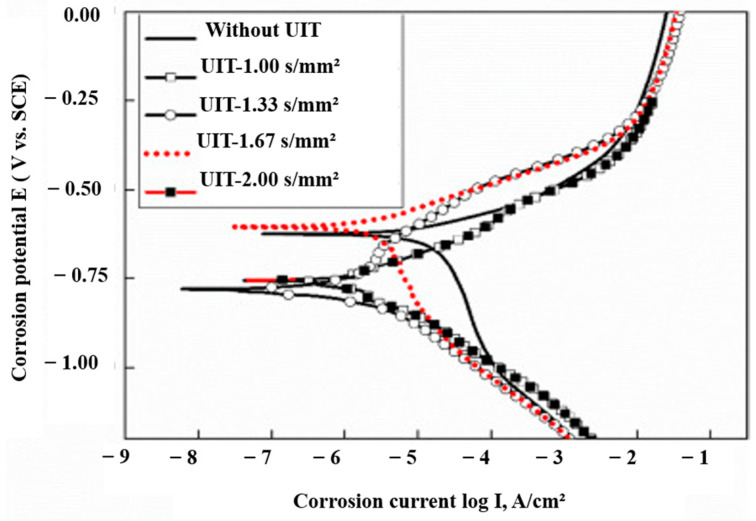
Polarization curves for specimens with different impact intensities. Reproduced with permission from [92]. Copyright Elsevier, 2021.

**Figure 11 materials-14-03841-f011:**
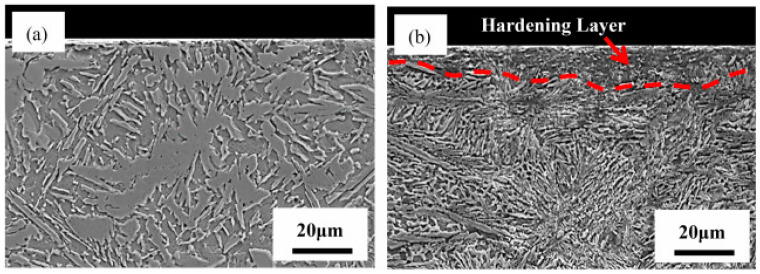
Cross-sectional morphologies of laser cladded AlCoCrCuFeNi HEA coatings: (**a**) without UIT; (**b**) with UIT. Reproduced with permission from [94]. Copyright Elsevier, 2021.

**Figure 12 materials-14-03841-f012:**
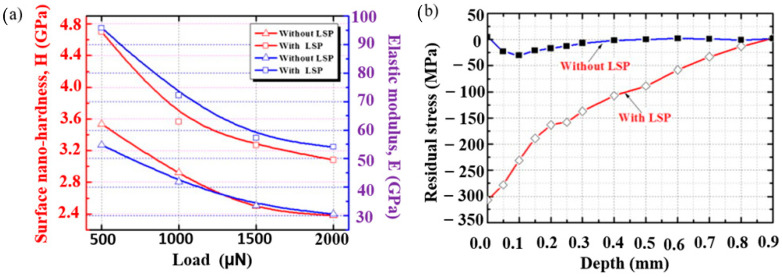
LSP on ANSI 304 SS and variation of (**a**) nano hardness and elastic moduli (**b**) residual stress with and without LSP. Reproduced with permission from [27]. Copyright Elsevier, 2011.

**Figure 13 materials-14-03841-f013:**
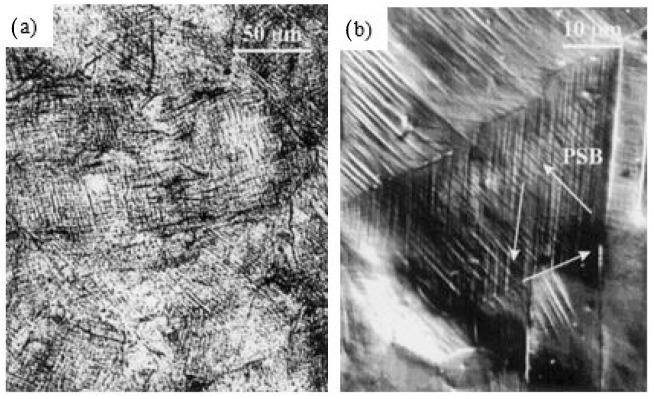
Optical micrographs of cross-section of 316L steel subjected to (**a**) SP; (**b**) LSP. Reproduced with permission from [115]. Copyright Elsevier, 2000.

**Figure 14 materials-14-03841-f014:**
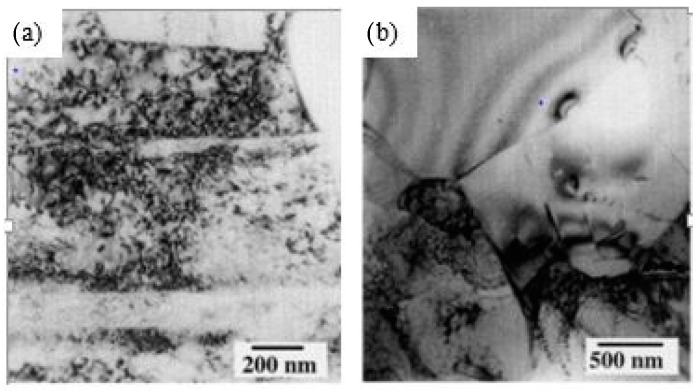
TEM on (**a**) LSP treated; (**b**) untreated specimen Ti6Al4V alloy. Reproduced with permission from [116]. Copyright Elsevier, 1999.

**Figure 15 materials-14-03841-f015:**
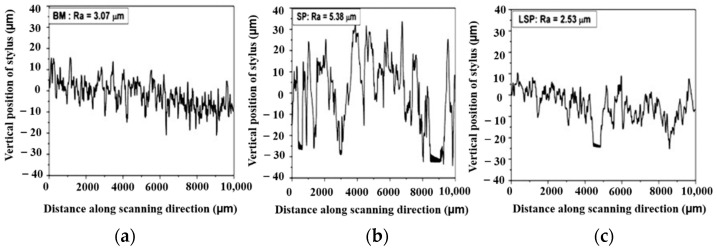
Surface profile comparisons (**a**) BM; (**b**) SP and (**c**) LSP treated specimen. Reproduced with permission from [117]. Copyright Elsevier, 2012.

**Figure 16 materials-14-03841-f016:**
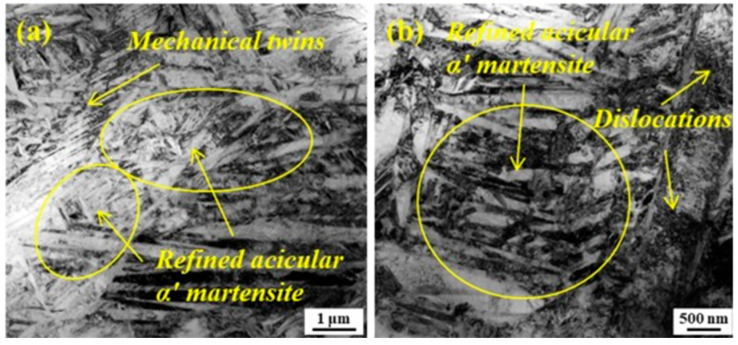
Typical TEM images of the LSP’ed SLM specimen. (**a**) Refined acicular martensite and MTs (**b**) dislocations and refined acicular martensite. Reproduced with permission from [120]. Copyright Elsevier, 2021.

**Figure 17 materials-14-03841-f017:**
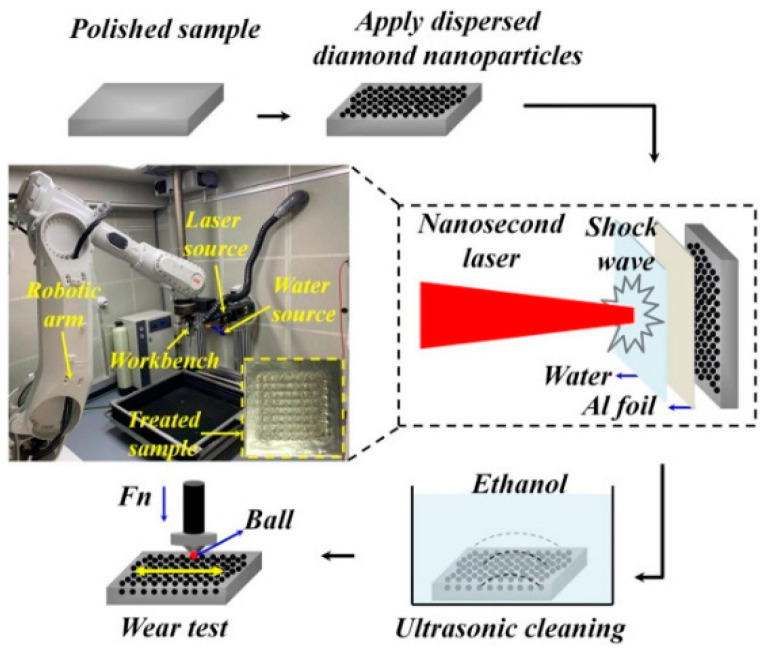
The experimental system of nanoparticle implantation induced by LSP. Reproduced with permission from [123]. Copyright Elsevier, 2021.

**Figure 18 materials-14-03841-f018:**
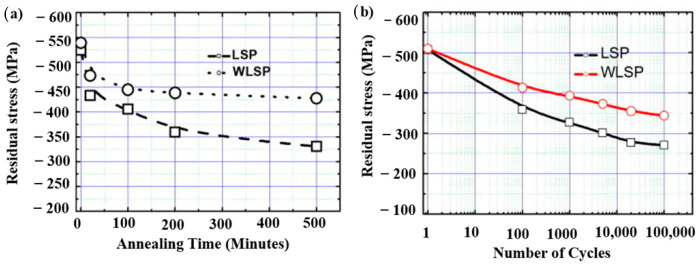
Comparison of WLSP and LSP on AISI 4140 steel (**a**) stress relaxation at 300 °C; (**b**) stress relaxation after cyclic loading. Reproduced with permission from [131]. Copyright Elsevier, 2011.

**Figure 19 materials-14-03841-f019:**
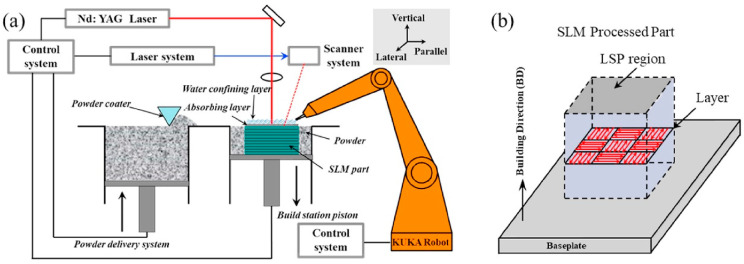
Integrated SLM with LSP (**a**) Process diagram and (**b**) laser scanning strategy. Reproduced with permission from [136]. Copyright Elsevier, 2020.

**Figure 20 materials-14-03841-f020:**
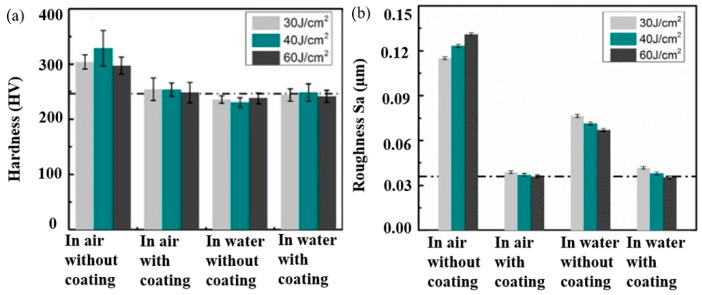
(**a**) Surface hardness and (**b**) roughness measurements in fs-LSP processed SS 304 under different conditions. Reproduced with permission from [140]. Copyright Elsevier, 2021.

**Figure 21 materials-14-03841-f021:**
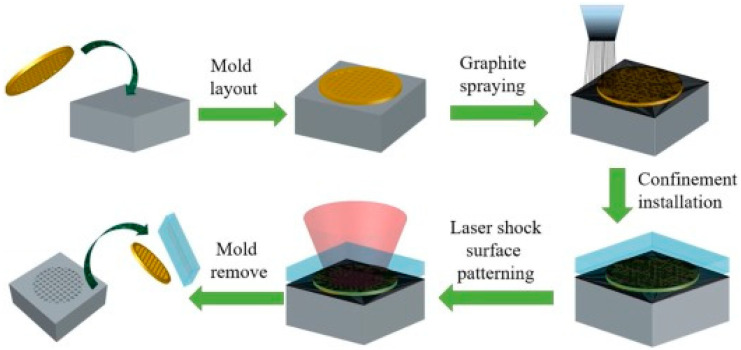
A schematic illustration of indirect-LSSP process. Reproduced with permission from [146]. Copyright Elsevier, 2018.

**Figure 22 materials-14-03841-f022:**
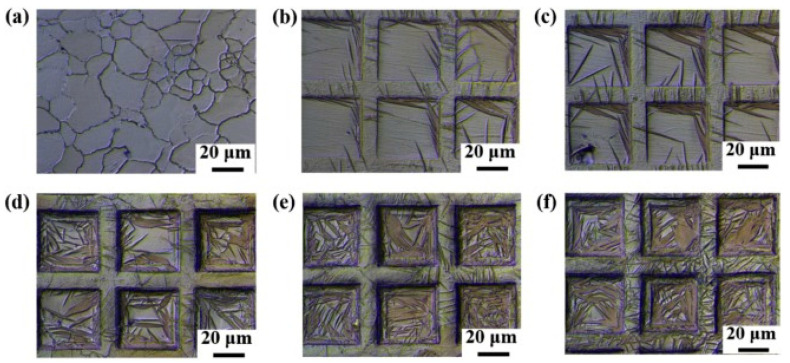
OM showing the microstructure of direct-LSSP-patterned AZ31B with laser intensities of: (**a**) 0; (**b**) 1.18; (**c**) 1.47; (**d**) 1.70; (**e**) 1.92; and (**f**) 2.12 GW/cm^2^. Reproduced with permission from [147]. Copyright Elsevier, 2020.

**Figure 23 materials-14-03841-f023:**
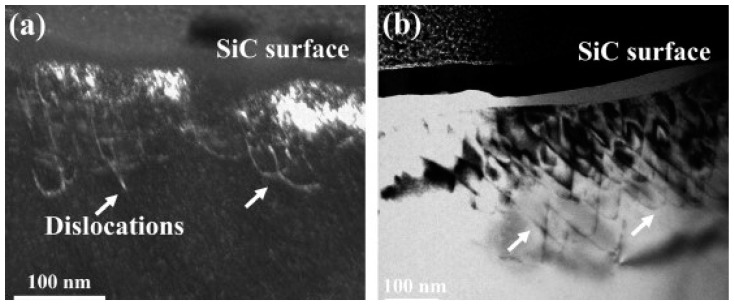
TEM images of dislocations in *α*-SiC ceramics generated by LSP: (**a**) weak-beam dark- field and (**b**) bright-field images of dislocations underneath the surface. Reproduced with permission from [149]. Copyright Elsevier, 2019.

**Table 1 materials-14-03841-t001:** Effect of classical and advanced SP techniques on engineering materials.

Substrate	Technique	Findings	References
AISI 304 SS	SP	Microhardness increased by 52%, strength by 14% and fracture toughness by 18%.	[33]
Ti6Al4V	SP	Improved the fatigue life by 34% and reduced the short crack propagation rate by 34–60% compared to unpeened specimen.	[34]
AISI 4140 low alloy steel	SP	Increased corrosion resistance, surface roughness, promoted grain refinement and subgrain formation. The corrosion mechanism changed from uniform corrosion to crevice corrosion	[57]
hastelloy X alloy	SSP	After SSP, residual stress at depth 125 μm is 1200 MPa, average grain size on the surface ~50 nm, depth of compressed layer was 700 μm and hardness on the surface 2.2 times compared to SP	[64]
X70 steel	SSP	Increased fatigue performance Improved work hardening and surface roughness	[65]
AZ31 and AZ91 magnesium alloys	SSP	Nano grains on deformed layer Microhardness of both alloys increased The corrosion resistance of AZ31 alloy improved	[66]
Pure Ti	SSP	Ultra-fine grained surface with grain size 100 nm Microhardness and elastic moduli increased	[67]
Cast iron	SSP & RSSP	High work hardening, deeper RCS, nano crystallization RP reduced surface roughness and improved fatigue performance.	[71]
High speed tool steel	MSP	Residual stress was higher on the surface with low surface roughness Improved peening effect	[14]
Structural steel	MSP	Improve fatigue performance and wear resistance Enhanced peening effect	[15]

**Table 2 materials-14-03841-t002:** Effect of UIP process on various engineering materials.

Substrate	Findings	References
High-nitrogen austenitic SS	Fatigue life enhanced at low strain amplitude by 18% Grain size of 15 and 12 nm observed for peening duration of 3 and 18 min The thickness of the refined region is approximately 260 μm and 345 μm for 3 and 18 min peening	[87]
AZ31 Magnesium alloy	The grain size on the surface is 37 nm Microhardness at the surface increased by 141% Coefficient of friction reduced Improved wear resistance Delamination wear mechanism retarded after UIP	[88]
7075 Aluminum alloy	Observed surface nanocrystalization Refined grains Improved corrosion resistance	[89]
7150 Aluminum alloy	Observed surface nanocrystalization Exfoliation susceptibility decreased Corrosion resistance increased	[90]
β-titanium alloy	A nanocrystalline layer of 100 μm thickness on the surface Improved microhardness No new phase formed, decreased β phase because of stress-induced martensite	[91]

**Table 3 materials-14-03841-t003:** Typical process parameters of LSP and findings from the literature.

Material	Findings	References
ANSI 316L SS	Surface hardness improved by 35% Improvement in corrosion resistance	[20]
7075 Aluminium alloy	Hardness increased Abrasion resistance improved	[21]
Duplex SS	Wear volume reduced by 39% Corrosion rate reduced by 74.2% Corrosion pit size reduced by 50%	[22]
Ti-17	Fatigue life increased Microhardness increased Grain refinement	[25]
Alloy 718	Observed nanocrystallites and grain refinement at the surface Surface hardness increased Fretting wear resistance increased	[26]
ANSI 304 austenitic SS	Nano hardness improved Elastic modulus increased High RCS observed Mechanical twin formation observed	[27]
AZ31B magnesium alloy	Hardness increased by 20% Yield strength increased by 18.75% Refined grains Improved wear resistance	[28]
Polycrystalline α-Al2O3 Ceramics	Improved resistance to indentation cracking. Plastic deformation occurred at the grain boundary and elastic deformation in α-Al₂O₃ grains	[29]
Alloy D9	Microhardness increased by 32% Yield strength increased by 63% Improved thermal stability	[30]
Brass H62	Surface roughness increased by 28.3% Wear mass loss decreased by 31.78%	[31]

**Table 4 materials-14-03841-t004:** Effect of LSP on various weld joints.

Materials	Type of Welding	Remarks	References
ANSI 304 SS	LBW	Surface roughness in weld zone (WZ) and heat affected zone (HAZ) reduced Significant residual compressive stress in weldment Refined grains in WZ and HAZ	[107]
Alloy 600	TIG	Tensile strength of joint increased by 9% Yield of joint increased by 25% Improvement in microhardness, dislocation density Improved fatigue resistance of joint	[111]
Inconel 600	ATIG	Tensile fracture location changed from weld to base material side Weldment tensile strength and hardness value increased	[113]
7050-T7451 aluminum alloys	FSW	Hardness in TMAZ and HAZ increased Fatigue life increased by 30%, 27%, and 5% under different loading conditions	[114]

**Table 5 materials-14-03841-t005:** Recent developments of LSP.

LSP Techniques	Applications	References
LPwC	Used where sacrificial coating is difficult to apply	[124,125]
WLSP	Precipitate hardenable materials	[129,130]
CLSP	Metals that form deformation twins	[133]
fs-LSP	Circumstances where confining medium and sacrificial coating is difficult to apply	[140]
LPF	For shaping and forming components with complex shapes	[141,142,143]
EP-LSP	Low plasticity materials	[148]

## Data Availability

Data sharing is not applicable to this article.

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
