# Peer review of "Peening Techniques for Surface Modification: Processes, Properties, and Applications"

_materials, 2021, doi:10.3390/ma14143841_

Round 1
Reviewer 1 Report
The authors present a review on a material surface modification method using peening techniques. The paper is well written. The current state-of-the-art of peening technologies is extensively discussed. Following comments are given.
1. The paper addresses peening techniques for surface modification. Then the title of the paper should be modified.
2. Some part of the introduction section can be moved to the main section. In the introduction, the authors need to mention the main contents to be presented in the paper and how the paper is organized.
3. In the main section, each sub-section should be better categorized. It makes easier for the reader to distinguish.
4. Error in reference sources in many pages.
Author Response
Please find the attached rebuttal for Reviewer 1 comments

Reviewer 2 Report
The authors have done a literature survey on the state-of-the-art of laser assisted surface modification. The manuscript reads well and covers different aspects of the peening technique. However, there are minor issues that needs to be address before publication.
The reviewed subjects do not reflect on the title, please chose a better title.
It would be useful if the author could provide more illustrative examples for each method.
The authors may want to include the residual stress induce onto the Si substrate using LSP for biological application, in “Applications and future directions”. i.e., doi.org/10.1038/srep35425
Author Response
Please find the attached rebuttal for Reviewer 2 comments

Reviewer 3 Report
Fig.4 it will be good to show or mark somehow (with arrows or other) where exactly is emission source of dislocations identified as Al2Cu and where are dislocation cells and tangling
Fig 5. How to understand the legend and views relantionship? If view a) is before fatigue experiment and b) is after, with microcracks, so where is modified curvature and crack elimination in peened specimen? Or it needs to be read in other way?
The advantage of article is, that list of references is quit large, lot of information was analysed. The review looks like nice compact summary of scientific works. But in my point of view there is a critical estimation from authors is missing.
And yes, I agree with the last sentence of authors, citation: "This review article can offer better insights when choosing the peening techniques for various industrial applications.". I see the article like good short handbook of peening techniques.
Author Response
Please find the attached rebuttal for Reviewer 3 comments
